# Glucosamine Enhancement of Learning and Memory Functions by Promoting Fibroblast Growth Factor 21 Production

**DOI:** 10.3390/ijms25084211

**Published:** 2024-04-10

**Authors:** Yu-Ming Chao, Hon-Yen Wu, Sin-Huei Yeh, Ding-I Yang, Lu-Shiun Her, Yuh-Lin Wu

**Affiliations:** 1Institute of Physiology, College of Medicine, National Yang Ming Chiao Tung University, Taipei 112304, Taiwan; chao.s976827@gmail.com (Y.-M.C.); au04294jo6vul4@gmail.com (S.-H.Y.); 2Division of Nephrology, Department of Internal Medicine, Far Eastern Memorial Hospital, New Taipei City 220, Taiwan; honyenwu@gmail.com; 3School of Medicine, College of Medicine, National Yang Ming Chiao Tung University, Taipei 112, Taiwan; 4Institute of Brain Science, College of Medicine, National Yang Ming Chiao Tung University, Taipei 112, Taiwan; diyang@nycu.edu.tw; 5Department of Life Sciences, College of Bioscience and Biotechnology, National Cheng Kung University, Tainan 701, Taiwan; lsher@mail.ncku.edu.tw

**Keywords:** glucosamine, learning and memory, fibroblast growth factor 21, signal transduction

## Abstract

Fibroblast growth factor 21 (FGF21) plays a crucial role in metabolism and brain function. Glucosamine (GLN) has been recognized for its diverse beneficial effects. This study aimed to elucidate the modulation of FGF21 production by GLN and its impact on learning and memory functions. Using both in vivo and in vitro models, we investigated the effects of GLN on mice fed with a normal diet or high-fat diet and on mouse HT22 hippocampal cells, STHdh^Q7/Q7^ striatal cells, and rat primary cortical neurons challenged with GLN. Our results indicated that GLN promotes learning and memory functions in mice and upregulates FGF21 expression in the hippocampus, cortex, and striatum, as well as in HT22 cells, STHdh^Q7/Q7^ cells, and cortical neurons. In animals receiving GLN together with an FGF21 receptor FGFR1 inhibitor (PD173074), the GLN-enhanced learning and memory functions and induction of FGF21 production in the hippocampus were significantly attenuated. While exploring the underlying molecular mechanisms, the potential involvement of NF-κB, Akt, p38, JNK, PKA, and PPARα in HT22 and NF-κB, Akt, p38, and PPARα in STHdh^Q7/Q7^ were noted; GLN was able to mediate the activation of p65, Akt, p38, and CREB in HT22 and p65, Akt, and p38 in STHdh^Q7/Q7^ cells. Our accumulated findings suggest that GLN may increase learning and memory functions by inducing FGF21 production in the brain. This induction appears to be mediated, at least in part, through GLN’s activation of the NF-κB, Akt, p38, and PKA/CREB pathways.

## 1. Introduction

Learning and memory functions are essential for individuals to engage with their environment and involve intricate communication within extensive brain networks [1]. Various brain regions, including the hippocampus, cortex, striatum, and amygdala, play important roles in consolidating different forms of learning and memory [2]. Indeed, the preservation of learning and memory is critical for one’s quality of life, and their loss or degeneration due to aging or diseases has emerged as a significant global health issue [3]. Previous studies have suggested the potential significance of fibroblast growth factor 21 (FGF21) and glucosamine (GLN) in facilitating learning and memory functions [4,5], suggesting that these novel molecules might benefit patients suffering from the degeneration of their learning and memory functions.

FGF21 has been recognized as a key physiological molecule in metabolism [6], with the liver being its primary production site [7]. Notably, FGF21 has been detected in diverse brain regions, such as the substantia nigra, hippocampus, cortex, striatum, and glial cells [8,9]. Moreover, the ability of plasma-derived FGF21 from peripheral tissues to cross the blood–brain barrier (BBB) has been reported [10], and the abundant expression of FGF receptors (FGFRs), including FGFR1, the preferential cell surface receptor for FGF21, has been reported [11], emphasizing the potential significance of FGF21 in brain physiology [12]. The potential neuroprotective roles of FGF21 have been demonstrated with in vivo and in vitro models [13,14,15]. In fact, FGF21 treatment has been suggested as an option for improving cognitive decline associated with neurodegeneration [16,17]. Diverse scenarios of action by FGF21 have been shown to confer its neuroprotective impact, such as by maintaining mitochondrial homeostasis [14,16], upregulating antioxidant activity [15,16], reducing neuroinflammation [15,17], and enhancing amyloid beta (Aβ) degradation [17].

As an essential component in glycoproteins and proteoglycans that are crucial for the formation of connective tissues and cartilage [18], GLN is a widely used dietary supplement that is recognized for its role in adjuvant therapy against osteoarthritis [19]. Aside from its well-established impact on cartilage preservation by inhibiting catabolic enzymes [20,21], GLN exhibits diverse effects, including anti-tumor [22], anti-oxidative [23], anti-allergic [24], and anti-inflammatory [25] properties. Notably, GLN’s ability to cross the BBB [26] suggests potential protective effects in the brain. In fact, GLN has been demonstrated to potentially act against pathologic conditions such as encephalomyelitis [27], learning and memory impairment [5], neuroinflammation, and ischemic brain injury [28,29,30].

Concerning the impacts of GLN and FGF21 on glucose and glycogen levels, prior research conducted on GLN-treated isolated rat adipocytes demonstrated a swift increase in glycogen biosynthesis and a reduction in glucose levels within minutes. However, prolonged exposure to high concentrations of GLN resulted in glucose accumulation and diminished glycogen biosynthesis, showing a biphasic effect of GLN. This variability may account for conflicting findings across different in vitro studies regarding the role of GLN in glucose/glycogen metabolism [31]. In human studies, while some investigations suggested potential GLN-mediated elevations in fasting glucose levels and reduced insulin sensitivity [32], a study involving 3063 human subjects found a slight decrease in fasting plasma glucose values after oral GLN treatment for 66 weeks [33]. Moreover, over a 3-year span of clinical trials encompassing healthy individuals and those with type 2 diabetes or prediabetes, GLN exhibited no significant impacts on fasting blood glucose levels, glucose metabolism, or insulin sensitivity across all three groups [34]. In contrast, studies in mice have indicated that recombinant FGF21 treatment lowers plasma glucose levels by enhancing glucose absorption and decreasing hepatic glucose release [35]. In obese mice, FGF21 treatment has been shown to notably enhance insulin sensitivity, promote energy expenditure, and augment fat oxidation while also inhibiting de novo lipogenesis in the liver, thereby ameliorating obesity and its associated metabolic complications [36,37]. Conversely, GLN administration in chow-diet-fed mice has been linked to increased body weight (BW) gain and white adipose tissue mass, alongside impaired insulin response in the liver. However, GLN has also demonstrated contradictory effects, such as reducing BW gain and liver weight and mitigating obesity-induced insulin resistance and impaired insulin signaling in the liver [38]. Currently, several studies consider FGF21 a promising target for treating obesity and metabolic diseases [39]. However, the potential therapeutic value of GLN in metabolism-related disorders remains uncertain, necessitating further evidence to substantiate its efficacy.

Our previous studies revealed GLN’s positive influence on cognitive performance [40] and its ability to induce FGF21 production in the liver and adipose tissues of mice [41]. Additionally, prior studies strongly suggested a neuroprotective role of FGF21 against cognitive degeneration [42,43]. However, whether GLN affects cognitive function by modulating FGF21 production remains an unsolved puzzle. Therefore, this study aimed to elucidate whether GLN would impact FGF21 production in brain tissues and nerve cells, to decipher the involvements of signaling components in GLN-induced FGF21 production, and to clarify the relation of GLN-mediated FGF21 upregulation to learning and memory functions by employing animal models and cell culture systems. Revealing a clear picture of how GLN-induced FGF21 production enhances learning and memory functions may illuminate a potential nondrug treatment for cognitive impairment.

## 2. Results

To test our hypothesis that GLN would upregulate FGF21 in brain tissues and, therefore, contribute to the enhancement of learning and memory functions, a mouse model was used to evaluate the learning and memory functions of mice and the associated FGF21 expression in critical brain tissues.

### 2.1. GLN’s Enhancement of Learning and Memory Functions in Normal Diet (ND)- and High-Fat Diet (HFD)-Fed Mice

When comparing the impacts of GLN in normal and obese animals throughout the 10-week feeding period with ND or HFD, the BW of HFD mice was found to be significantly higher than that of ND mice (Appendix A). The GLN treatment did not influence BW within the ND or HFD group. Interestingly, at various doses, the GLN treatment significantly enhanced learning and memory functions in both ND (3, 10, 30 mg/mouse) and HFD (10, 30 mg/mouse) animals (Figure 1A). There appeared to be no difference between the control animals in the ND and HFD groups (Figure 1A). Notably, the GLN treatment did not affect the total distance of animal movement within either the ND or HFD group (Figure 1B). No significant changes were observed in the total exploration times within the ND or HFD group (Appendix A). Moreover, both male and female animals receiving GLN demonstrated similar enhancements in learning and memory functions (Appendix A).

### 2.2. GLN’s Induction of FGF21 Expression in Brain Tissues and Nerve Cells

We previously showed that GLN treatment can mediate a cognition-enhancing function accompanied by an upregulation of BDNF levels in the hippocampus, cortex, and striatum [40]. Given that FGF21 has been reported to play a beneficial role in cognitive function [16,17], we looked into GLN’s potential regulation of FGF21 in the hippocampus, cortex, and striatum. GLN induced FGF21 production in all three tissues from both ND and HFD animals (Figure 2). In HT22 mouse hippocampal cells and STHdh^Q7/Q7^ mouse striatal cells, GLN (10, 20 mM) upregulated both the mRNA and protein levels of FGF21, and GLN (2.5, 5, 10, 20 mM) induced FGF21 secretion (Figure 3). Primary rat cortical neurons also exhibited an induction of FGF21 production upon exposure to GLN (Appendix A). Importantly, the GLN doses used in these cell types did not appear to affect cell viability (Appendix A).

### 2.3. Potential Involvement of FGF21 in the GLN-induced Enhancement of Learning and Memory Functions

To assess the potential involvement of FGF21 in the GLN-induced enhancement of learning and memory functions, the FGFR1 inhibitor PD173074 [44] was co-administered with GLN (30 mg/mouse). PD173074 attenuated the GLN-mediated enhancement (Figure 4A), with no impact on the total moving distance (Appendix A). Interestingly, PD173074 also suppressed GLN-induced FGF21 production in the hippocampus (Figure 4B). In HT22 or STHdh^Q7/Q7^ cells exposed to GLN (20 mM) in combination with PD173074, PD173074 (1 μM) partially reduced the FGF21 production induced by GLN (Figure 4C).

### 2.4. Identification of the Potential Target Signaling Molecules of GLN for Regulating FGF21 Production

To explore the signaling pathways involved in GLN-induced FGF21 expression and given that under the different physiological conditions in different cell types, previous studies demonstrated involvements of a variety of signal pathways in FGF21 expression, at least including NF-κB [41], cAMP/PKA and MAPK p38 [41], Akt [41,45], and PPARα and PPARγ [46], selective inhibitors against these diverse signaling molecules were employed. In HT22 cells, inhibitors for NF-κB, p38, JNK, Akt, PKA, and PPARα suppressed GLN-induced FGF21 production (Figure 5A). In STHdh^Q7/Q7^ cells, inhibitors for NF-κB, p38, Akt, PKA, and PPARα also reduced GLN-induced FGF21 production (Figure 5B). Further analysis in HT22 cells revealed the rapid induction of p65 and Akt phosphorylation at 5 min, p38 phosphorylation at 60 min, and CREB phosphorylation at 24 h by GLN (Figure 6), while JNK phosphorylation was unaffected by GLN (Appendix A). In STHdh^Q7/Q7^ cells, GLN induced p65 phosphorylation at 5 min and p38 and Akt phosphorylation at 60 min (Appendix A). To monitor the effect of GLN on PPAR signaling, the HT22 or STHdh^Q7/Q7^ cells were transfected with a PPRE reporter plasmid, and GLN did not activate but reduced the PPRE reporter activity (Appendix A). To test the critical role of NF-κB in GLN-induced FGF21 expression, HT22 cells were transfected with either EGFP or EGFP-RelA (p65) overexpression plasmid and then treated with or without GLN for 24 h. The transfection of HT22 cells with EGFP-RelA (p65) overexpression plasmid and subsequent GLN treatment resulted in a more profound induction of FGF21 compared with that in the EGFP control group (Figure 7).

## 3. Discussion

Given the demonstrated neuroprotective role of FGF21 and its potential to ameliorate cognitive decline, as evidenced in previous studies [16,17], along with the reported induction of FGF21 by GLN [41], our present research unequivocally illustrates that GLN mediates FGF21 production and enhances learning and memory functions in both ND and HFD mice, with a significant contribution by FGF21. This enhancement is accompanied by increased FGF21 expression in critical brain regions, including the hippocampus, cortex, and striatum. Moreover, we observed a significant upregulation of cellular FGF21 synthesis by GLN in HT22 hippocampal cells, STHdh^Q7/Q7^ striatal cells, and primary cortical neurons, indicating a broader impact on neural tissues. The mechanism underlying this GLN-induced effect potentially involves the activation of the p65, Akt, p38, and PKA/CREB pathways. Our findings strongly suggest that FGF21 may function downstream of GLN to confer beneficial effects on learning and memory functions.

GLN has already been shown to promote cognitive performance in mice, which is accompanied by induced BDNF expression and cAMP/PKA pathway activation in brain tissues [40]. In addition, the contribution of GLN to the alleviation of various brain pathologic scenarios has been previously shown—it potentially acts by suppressing inflammatory cascades in animal models of autoimmune encephalomyelitis [27] and ischemia [28,30] and reducing the phosphorylation of the Tau protein in a brain ischemia-reperfusion model [29]. Differently from these studies, our current findings identified a novel molecule FGF21 induced by GLN as the key molecule in the brain, presumably playing beneficial roles in learning and memory functions.

Previous research suggested that a minimum daily intake of 1500 mg of GLN is recommended for humans to achieve a beneficial effect [47]. However, due to GLN’s relatively low bioavailability stemming from its high clearance rate [48], administering GLN at 10 and 20 mg per mouse (equivalent to 400 and 800 mg/kg of mouse with 25 g BW) via intraperitoneal (IP) injection has proven effective in reducing atopic dermatitis and inflammation [49]. Previous studies indicated that the LD_50_ of GLN in rats, mice, and rabbits exceeded 5000 mg/kg BW [33], affirming the safety of daily IP injection of GLN (3, 10, 30 mg/mouse, approximately 120 to 1200 mg/kg) as utilized in this study. Considering that GLN is primarily consumed orally by humans, future animal models should be orally administered varying doses of GLN to evaluate its effects on learning and memory functions.

In our current studies in ND or HFD animals, regardless of gender, the index of learning and memory functions was significantly higher in the GLN treatment group than in the control (Figure 1, Appendix A). Notably, the total exploring time and movement distance did not display significant differences between the GLN-treated and control groups (Figure 1, Appendix A), indicating that GLN did not affect motor ability. While it is well known that obesity impairs learning and memory functions in both humans and mice [50,51], interestingly, our studies did not reveal a significant reduction in learning and memory functions due to HFD (Figure 1). This observation might be attributed to the relatively short duration of the HFD in our study (10 weeks), whereas cognitive impairment induced by obesity in mice typically requires a longer duration (approximately 20–40 weeks) of the HFD regimen [10]. Interestingly, cognitive deficits have been observed in animals subjected to shorter durations of HFD treatment. For instance, a study conducted on male rats (120–150 g) exposed to an HFD (SDS 824053; kcal composition: 45% fat, 20% protein, 35% carbohydrate) for 10 weeks demonstrated cognitive impairment [52]. Similarly, another study utilizing male mice of the same strain (aged 4–5 weeks) and HFD (Research Diets D12492: 60% fat, 20% protein, 20% carbohydrate), as in our current study, found that compared with the ND and low-fat diet (LFD) animals, cognitive function was impaired after just 1 week of the HFD; however, it returned to levels comparable to those in LFD mice after 6 weeks of the HFD [53]. In our study, 8-week-old mice were fed with an HFD for 10 weeks before undergoing an NORT. The discrepancies between our study and these reports may be attributed to differences in species and ages during the HFD exposure.

The considerable enhancement in hippocampal synaptic plasticity and dendritic spine density by FGF21 [54], along with its protective effects against Alzheimer’s disease (AD)-like pathologies induced by Aβ25-35 peptide [55] and cognitive impairment induced by D-galactose and an HFD [56], led us to hypothesize that the GLN-induced increase in FGF21 may significantly contribute to the enhanced learning and memory functions in mice (Figure 1). Indeed, our findings demonstrating that the injection of the FGFR1 inhibitor PD173074 notably mitigated the promotive effect of GLN (Figure 4A) strongly suggest that GLN might act by stimulating the production of FGF21 to mediate such beneficial effects. Considering that IP injection of the FGFR1 inhibitor PD173074 at 1 mg/kg/day has been utilized to inhibit FGFs-induced angiogenesis [57] and at 20–50 mg/kg/day to elucidate the anti-tumor effect of FGFs [58], 2 mg/kg/day was administered in our study. Intriguingly, the FGFR1 inhibitor also attenuated GLN-induced FGF21 production in the hippocampus (Figure 4B), and in both HT22 and STHdh^Q7/Q7^ cells, the inhibition of FGFR1 reduced GLN-induced FGF21 production (Figure 4C), suggesting a potential autocrine or paracrine regulatory mechanism of FGF21 in the brain. While the liver is commonly regarded as the primary source of FGF21 production [6], our current research indicates that brain tissues, including the hippocampus, cortex, and striatum, also have the capacity to generate FGF21, a process further enhanced by GLN (Figure 2). Given that FGF21 has been noted to traverse the BBB [59] and that previous studies have demonstrated GLN’s ability to significantly boost FGF21 levels in liver and adipose tissues [41], it is conceivable that GLN-induced FGF21 production from these peripheral tissues may also contribute to the enhancement of learning and memory functions promoted by GLN. Consequently, further investigation is needed to elucidate whether GLN-induced FGF21 production in the brain, peripheral tissues, or both plays a significant role in mediating GLN’s effects on learning and memory functions. Additionally, although direct evidence regarding whether PD173074 can cross the BBB is lacking, a prior study administering PD173074 at 10 mg/kg in conjunction with basic FGF (bFGF) via IP injection showed a reduction in bFGF-mediated protection against traumatic brain injury in rats [60]. This suggests the potential for PD173074 to penetrate the BBB and effectively inhibit FGFR activity in the brain. Despite PD173074 being a potent FGFR1 inhibitor with an IC50 of 25 nM and demonstrating 1000-fold selectivity for FGFR1 over PDGFR and c-Src [61], it is important to consider that PD173074 treatment may not solely rely on FGFR1 blockade. Therefore, understanding the kinetics of PD173074 is critical for optimizing dosing regimens and predicting its efficacy, safety, and specificity profiles in future studies while elucidating the specific roles of FGF21.

In our in vitro experiments in mouse HT22 hippocampal cells, STHdh^Q7/Q7^ striatal cells, and rat primary cortical neurons, as well as in the animals’ hippocampus, cortex, and striatum, we consistently observed GLN’s induction of FGF21 production. While pinpointing the potential signal target molecules of GLN for inducing FGF21 production, we found similarities with previous studies that implicated the important roles of p38 and PKA [41,62,63], NF-κB [41], Akt [41,45], and PPARα and PPARγ [46]. However, there were notable differences between HT22 hippocampal cells and STHdh^Q7/Q7^ striatal cells in terms of the potential involvement of NF-κB, p38, JNK, Akt, PKA, and PPARα, as well as NF-κB, p38, Akt, and PPARα, in HT22 and STHdh^Q7/Q7^ cells, respectively (Figure 5). In addition to a previous report on the activation of NF-κB, p38, ERK, and PKA by GLN in mouse liver cells [41], the responsiveness to GLN in the activation of NF-κB, p38, Akt, and PKA/CREB in HT22 hippocampal cells (Figure 6) and NF-κB, p38, and Akt in STHdh^Q7/Q7^ striatal cells (Appendix A) suggests potential cell-type-specific regulatory mechanisms in GLN-mediated FGF21 expression. Notably, we are the first to demonstrate such diverse involvements of different signaling players in GLN-mediated FGF21 expression in different nerve cells. In line with previous discoveries highlighting PPARα’s role as a crucial mediator in FGF21 upregulation [46,64], our studies also revealed PPARα’s significance in the upregulation of FGF21 (Figure 5), despite GLN not directly activating the PPAR pathway (Appendix A). Presently, an intriguing unresolved question pertains to whether and how the NF-κB, p38, Akt, PKA, and PPARα pathways interact with each other to regulate FGF21 production in the brain, necessitating further investigation. Notably, a prior study in hepatocytes illustrated that p38 functions downstream of PKA in mediating glucagon-induced FGF21 production [63]. Hence, it would be valuable to explore potential cross-talk or feedback mechanisms to elucidate how these signaling components converge in mediating GLN-induced FGF21 production.

In both HT22 and STHdh^Q7/Q7^ cells, GLN induces the upregulation of both protein and mRNA levels of FGF21 (Figure 2), indicating a potential transcriptional regulation by GLN for the FGF21 gene. Notably, the presence of typical AP-1, which is often associated with MAPK signaling, and that of NF-κB-responsive elements are not detected in the FGF21 promoter sequences across human, mouse, or rat species [65]. This suggests that NF-κB and MAPK may not directly function downstream of GLN to activate FGF21 transcription. While the presence of PPARα elements has been confirmed in the FGF21 promoter region across all three species [65], our study indicates that GLN does not induce PPAR activation. Hence, further investigation is warranted to determine whether GLN affects the stability of FGF21 mRNA and/or protein, thereby potentially promoting FGF21 production.

Interestingly, there seems to be a trend of a dose-dependent effect of GLN on FGF21 expression in HT22 hippocampal and STHdh^Q7/Q7^ striatal cells (Figure 3). However, this dose-response effect appears to be less apparent in brain tissues (hippocampus, cortex, striatum) as it was shown that only the GLN dose at 30 mg/mouse resulted in significant induction of FGF21 in all three tissues of both ND- and HFD-fed animals (Figure 2). Similarly, while monitoring the learning and memory functions, GLN did not appear to give a typical dose-dependent enhancement (Figure 1). Therefore, these should warrant further investigation in the future to identify an optimal dose and duration of GLN treatment in animal models. Nevertheless, our current studies also indicated no significant difference between male and female mice in response to GLN treatment in terms of enhanced learning and memory functions. This finding points to an important issue in preclinical research regarding providing similar beneficial impacts between genders.

Our study has some limitations. First, while previous research in humans showed a correlation between increasing circulating FGF21 levels and the severity of obesity [66], and studies in male mice demonstrated elevated hepatic and serum FGF21 levels after exposure to an HFD for 14 or 16 weeks [67,68], this effect was not observed in female mice [67]. In our research, aside from BW, we did not assess various blood-based parameters to confirm obesity in all aspects or serum FGF21 levels. In our future studies involving obesity models, these factors should be taken into consideration. In addition, while involving a 10-week HFD regimen, we did not directly compare hepatic and serum FGF21 levels between the ND and HFD groups or between male and female mice. Additionally, although there is no existing literature on the impact of HFDs on GLN metabolism, GLN may be influenced by HFDs, warranting further investigation. Second, we only utilized the NORT to assess learning and memory functions. In future studies, additional monitoring protocols, such as the Morris water maze and Y-maze [69], should be employed to more accurately assess the effects of GLN on learning and memory functions in animals. Moreover, previous findings in mice showed that GLN can enhance cognitive function and induce BDNF expression [40], and our current study highlighted the role of FGF21 in the GLN-mediated enhancement of learning and memory functions. Exploring the potential interactions between FGF21 and BDNF under GLN regulation could provide further insights into their combined effects on learning and memory functions. Third, our study employed a panel of inhibitors targeting various signaling molecules. While each inhibitor is expected to selectively target a specific molecule, the possibility of off-target effects cannot be entirely disregarded. Therefore, in future investigations, a broader range of inhibitors and a series of dilutions thereof should be utilized for each target molecule. Additionally, employing combined knockdown and overexpression strategies to manipulate individual signaling molecules will be pursued to validate the findings obtained from inhibitor usage. As FGF21 has been proposed as a potential therapeutic agent for AD and Parkinson’s disease (PD) [12,70], investigating whether GLN-mediated FGF21 production exerts beneficial effects in animal models of AD and PD could inform potential therapeutic strategies for neurodegenerative diseases or cognitive impairment in the future. Furthermore, GLN, which has been identified as either a food additive or a nutraceutical, is typically taken orally. Studies have indicated a low bioavailability of GLN at around 6% in animals [71]. Additionally, previous research involving osteoarthritis patients and normal individuals without medical conditions who received oral GLN showed significant variability in steady-state plasma concentrations. This suggests notable differences among individuals in how GLN is absorbed and eliminated [72,73]. Consequently, to develop therapies targeting FGF21 production induced by GLN for cognitive disorders, enhancing GLN’s bioavailability through dosage optimization and alternative administration methods is imperative to maximize its effects.

In conclusion, our research presents compelling evidence of GLN’s capacity to promote FGF21 production in key brain regions, thereby enhancing learning and memory functions. GLN achieves this by upregulating mRNA and protein levels, as well as by promoting the secretion of FGF21 in hippocampal and striatal cells. These effects are mediated, at least in part, through the activation of several signaling pathways, including those of NF-κB, p38, Akt, and PKA. Our findings illuminate the potential cognitive advantages of GLN by modulating FGF21 production and offer valuable insights into the molecular mechanisms underlying GLN’s regulation of FGF21. Considering the significance of dietary supplements, various natural compounds and medicinal plants have demonstrated beneficial effects on neurological disorders and brain health [74]. Indeed, the combination of GLN with nonsteroidal anti-inflammatory drugs for treating inflammatory conditions appears promising as it not only enhances efficacy beyond individual action but also shows no significant evidence of drug interactions [75]. To develop innovative treatment strategies for neural diseases, further exploration is warranted to investigate whether GLN can effectively collaborate with existing medications to exert more potent disease-modifying or symptom-alleviating effects in cognitive disorders, thus necessitating extensive future research.

## 4. Materials and Methods

### 4.1. Chemicals and Reagents

Fetal bovine serum (FBS) was purchased from Cytiva (Marlborough, MA, USA). Reverse transcriptase and SYBR green reagents were obtained from ThermoFisher (ThermoFisher Scientific, Waltham, MA, USA). Antibodies were obtained from different sources: anti-β-actin (Novus Cat. # NB600-501; St. Louis, MO, USA), anti-FGF21 (abcam Cat. #ab171941), anti-phospho-Akt (Ser473) (Cell Signaling, Cat. # 9271; Danvers, MA, USA*)*, anti-Akt (Cell Signaling, Cat. # 4691), anti-phospho-CREB (87G3) (Cell Signaling, Cat. # 9197), anti-CREB (48H2) (Cell Signaling Cat. # 9197), anti-phospho-JNK (Thr183/Thr185) (Cell Signaling, Cat. # 9251), anti-JNK (Cell Signaling, Cat. # 9258), anti-phospho-p38 (Thr182) (Santa Cruz Cat. # sc-7973), anti-p38 (Cell Signaling, Cat. # 9212), anti-phospho-NF-κB p65 (S536) (R&D Systems, Cat. # KCB7226; Minneapolis, MN, USA), and anti-NF-κB p65 (Cell Signaling, Cat. # 8242). The materials for the animal diets were obtained from Research Diets Inc. (New Brunswick, NJ, USA). Glucosamine hydrochloride (Cat. 66842), along with all other unspecified chemicals and reagents utilized in this project, was procured from Sigma-Aldrich (St. Louis, MO, USA).

### 4.2. Animal Ethics

Male C57BL/6 mice aged seven weeks were obtained from the National Laboratory Animal Center in Taiwan. The animals were housed under controlled conditions of humidity, temperature, and lighting. All procedures involving animals were ethically approved by the Institutional Animal Care and Use Committee of the National Yang Ming Chiao Tung University (Permit Number 1100413).

### 4.3. Animal Experiments

The animals were given a one-week acclimation period in standard environmental conditions prior to experimentation. Mice were then assigned to either an ND (LabDiet 5001: 11.5% fat, 24.1% protein, 64.4% carbohydrate) or an HFD (Research Diets D12492: 60% fat, 20% protein, 20% carbohydrate) for a duration of 10 weeks. Starting from week 8 and continuing through week 10, the animals received daily IP injections containing varying doses (0, 3, 10, 30 mg/mouse) of GLN or GLN (0, 30 mg/mouse) in combination with PD173074 (2 mg/kg). Although the potential of anti-FGFR therapy in relation to FGF21 in cancer treatment has been acknowledged, extensive clinical trials on PD173074 are yet to be conducted [76]. While specific information regarding the in vivo pharmacokinetics of PD173074 is limited, various studies have explored its pretreatment [77] or cotreatment [78] with other agents in animal models to elucidate the crucial roles of FGF21 in different systems. Moreover, IP administration of PD173074 has been commonly utilized in animal studies [79,80]. Assessment of learning and memory functions was conducted using the NORT [81], which consisted of habituation, training, and testing phases. For the habituation phase, the mice were transferred into an empty cage to adapt to the environment for 15 min. On the next day, the mice were transferred to the cage again with the presence of two identical objects located at two corners. The mice were allowed to explore the objects for 15 min. For the testing phase, one of the objects was replaced with a novel object. The mice were allowed to explore the objects for 10 min, and the time spent exploring the familiar object (tA) and novel object (tB) was recorded. Their learning and memory functions were evaluated with the recognition index (RI) = tB/(TA + tB), followed by statistical analysis. The movement of the mice was recorded and analyzed using SMART VIDEO TRACKING software 3.0. Following experimentation, animals were euthanized via cervical dislocation, and their brains were dissected into the hippocampus, cortex, and striatum for further analysis.

### 4.4. Cell Culture

The HT22 mouse hippocampal neuronal cell line (passage 5–11) [82] and STHdh^Q7/Q7^ mouse striatal cell line (passage 5–10) [83] were cultured in DMEM–high glucose supplemented with 10% fetal bovine serum, 100 units/mL penicillin, and 100 μg/mL streptomycin.

### 4.5. Western Blotting Analysis

Total cellular proteins were extracted and quantified using the Bio-Rad protein assay reagent (Bio-Rad, Hercules, CA, USA), and the total protein concentration was adjusted with SDS-PAGE loading buffer and heated to 100 °C for 10 min and then subjected to a regular Western blotting assay to determine the expression profiles of various proteins. Equal amounts (50 μg) of proteins of samples were separated with 10% SDS-PAGE, transferred onto nitrocellulose membranes, blocked with 5% milk, and then incubated overnight with specific primary antibodies. Following primary antibody incubation, membranes were probed with corresponding horseradish-peroxidase-coupled secondary antibodies. The membrane was exposed to film, and the protein bands were visualized and quantified using the ImageQuant 5.2 software (Molecular Dynamics, Sunnyvale, CA, USA).

### 4.6. Enzyme-Linked Immunosorbent Assay (ELISA)

The concentration of FGF21 in the culture medium was determined using an ELISA kit for mouse FGF21 (R&D systems, Minneapolis, MN, USA) according to the manufacturer’s instructions.

### 4.7. Quantitative Reverse Transcription Polymerase Chain Reaction (Q-RT-PCR)

Total cellular RNAs were extracted from the harvested tissues and treated HT22 or STHdh^Q7/Q7^ cells using Tri-reagent according to the manufacturer’s protocol (Sigma). The RNA samples were resuspended in RNase-free diethyl-pyrocarbonate-treated water, and then each sample underwent quantitative RT-PCR to quantify the levels of mRNAs using specific primer sequences: mouse FGF21 (sense: 5′-CAC CGC AGT CCA GAA AGT CT-3′, antisense: 5′-ATC CTG GTT TGG GGA GTC CT-3′) yielding a 246 bp product; mouse GAPDH (sense: 5′-AAG GTC ATC CCA GAG CTG AA-3′, antisense: 5′-CTG CTT CAC CAC CTT CTT GA-3′) yielding a 222 bp product.

### 4.8. Statistical Analysis

The experimental data are expressed as the mean plus/minus the standard errors of the means (mean ± S.E.M.) for the indicated numbers of animals or repeated experiments. Statistical analysis was conducted using GraphPad Prism. For most comparisons, a one-way ANOVA with Tukey’s multiple-comparison test was employed. However, when comparing the ND and HFD groups, two-way ANOVA with Tukey’s multiple-comparison test was utilized. The *p*-values of <0.05 were considered statistically significant.

## Figures and Tables

**Figure 1 ijms-25-04211-f001:**
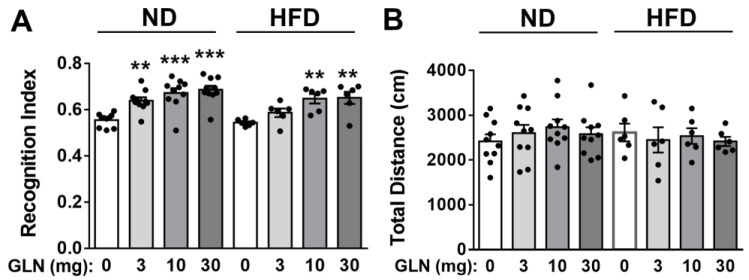
GLN’s impact on learning and memory functions in mice fed with an ND and HFD. During the 10-week dietary period with either an ND or HFD, mice were given GLN (0, 3, 10, or 30 mg/mouse/day) for 14 consecutive days, spanning from week 8 to week 10. A novel object recognition test (NORT) was employed to assess learning and memory functions by monitoring the time spent by the animals exploring a novel object. The recognition index (**A**) and total moving distance (**B**) were recorded for the ND mice and HFD mice. Data are presented as means ± S.E.M. (ND: *n* = 10; HFD: *n* = 6). ** *p* < 0.01 and *** *p* < 0.001 compared with GLN at 0 mg/mouse within the ND or HFD group.

**Figure 2 ijms-25-04211-f002:**
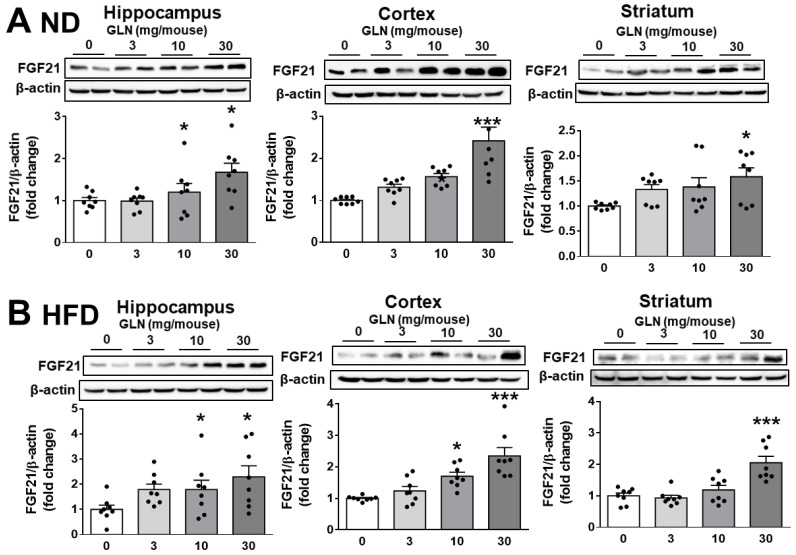
GLN’s induction of FGF21 expression in the hippocampus, cortex, and striatum. After two weeks of GLN injection, the protein levels of FGF21 were measured in the hippocampus, cortex, and striatum of ND (**A**) or HFD mice (**B**) using Western blotting with β-actin as an internal control. The results represent the means ± S.E.M. (*n* = 8). * *p* < 0.05 and *** *p* < 0.001 compared with the GLN 0 mg/mouse group.

**Figure 3 ijms-25-04211-f003:**
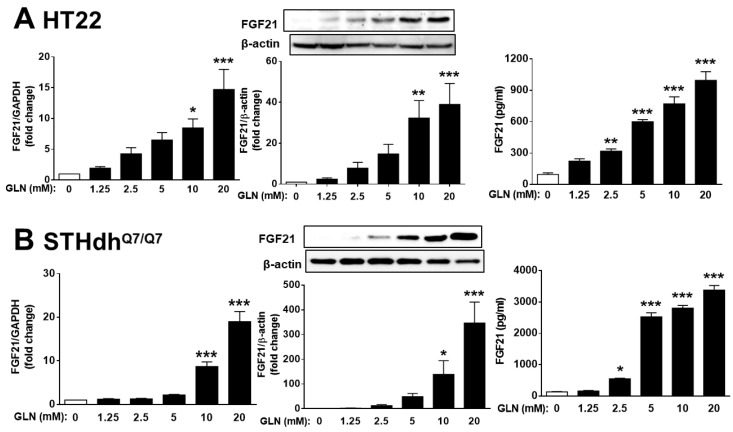
GLN-induced expression of FGF21 in hippocampal and striatal cells. HT22 hippocampal (**A**) and STHdh^Q7/Q7^ striatal cells (**B**) were treated with GLN (0, 1.25, 2.5, 5, 10, 20 mM) for 6 h for mRNA determination or 24 h for protein measurement. FGF21 mRNA was quantified using real-time PCR with GAPDH as an internal control, while FGF21 protein was examined through Western blotting with β-actin as an internal control. FGF21 production in the cultured medium was assessed with ELISA. The results represent the means ± S.E.M. (*n* = 4). * *p* < 0.05, ** *p* < 0.01, and *** *p* < 0.001 compared with the GLN 0 mM group.

**Figure 4 ijms-25-04211-f004:**
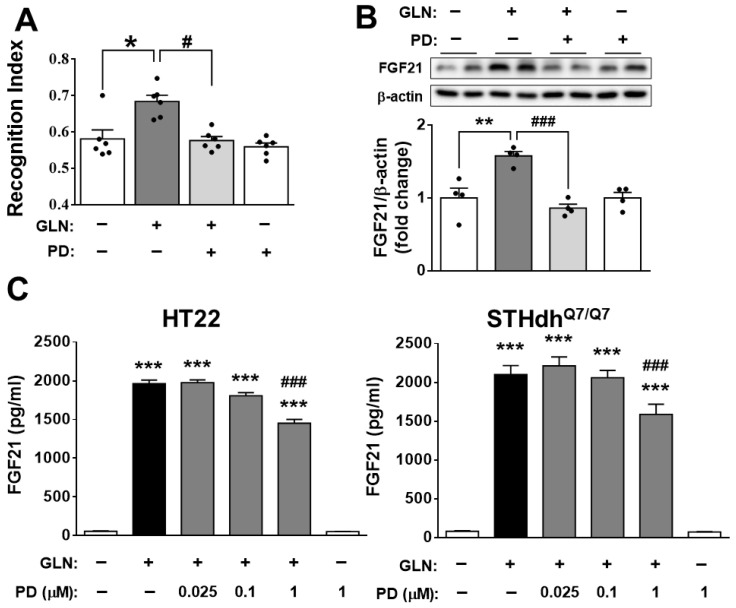
Attenuation of GLN-enhanced learning and memory functions and FGF21 production through FGFR1 inhibition in the hippocampus, hippocampal cells, and striatal cells. The ND-fed mice were administrated GLN (0, 30 mg/mouse) alone or in combination with PD173074 (2 mg/kg) through IP injection for 2 weeks, followed by the NORT (**A**). The FGF21 protein levels in the hippocampus (**B**) and in HT22 or STHdh^Q7/Q7^ cells treated with GLN (20 mM) alone or in combination with PD173074 (0.025, 0.1, 1 μM) for 24 h (**C**) were determined through Western blotting or ELISA. The results represent the means ± S.E.M. from 6 animals (**A**) or 4 animals (**B**) in each treatment group or from 4 independent experiments (**C**). * *p* < 0.05, ** *p* < 0.01, and *** *p* < 0.001 compared with the GLN 0 mg/mouse group or the GLN 0 mM group. ^#^ *p* < 0.05 and ^###^ *p* < 0.001 compared with the GLN 30 mg/mouse or the GLN 20 mM only group.

**Figure 5 ijms-25-04211-f005:**
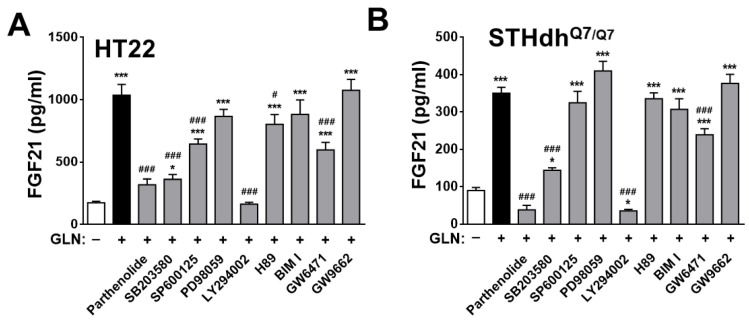
Illustration of the potential involvement of signaling molecules in GLN-mediated FGF21 production in hippocampal and striatal cells. (**A**) HT22 cells or (**B**) STHdh^Q7/Q7^ cells were pretreated with specific inhibitors targeting NF-κB (Parthenolide, 5 µM), MAPK p38 (SB203580, 20 µM), JNK (SP600125, 20 µM), ERK (PD98059, 20 µM), Akt (LY294002, 10 μM), PKA (H89, 5 µM), PKC (Bisindolylmaleimide, BIMI, 5 µM), PPARα (GW6471, 10 µM), or PPARγ (GW9662, 10 µM) for 1 h before the addition of GLN (20 mM) for an additional 24 h. The FGF21 concentration in the cultured medium was quantified using ELISA. The results represent the means ± S.E.M. (*n* = 4). * *p* < 0.05 and *** *p* < 0.001 compared with the 0 mM group. ^#^ *p* < 0.05 and ^###^ *p* < 0.001 compared with the GLN 20 mM group.

**Figure 6 ijms-25-04211-f006:**
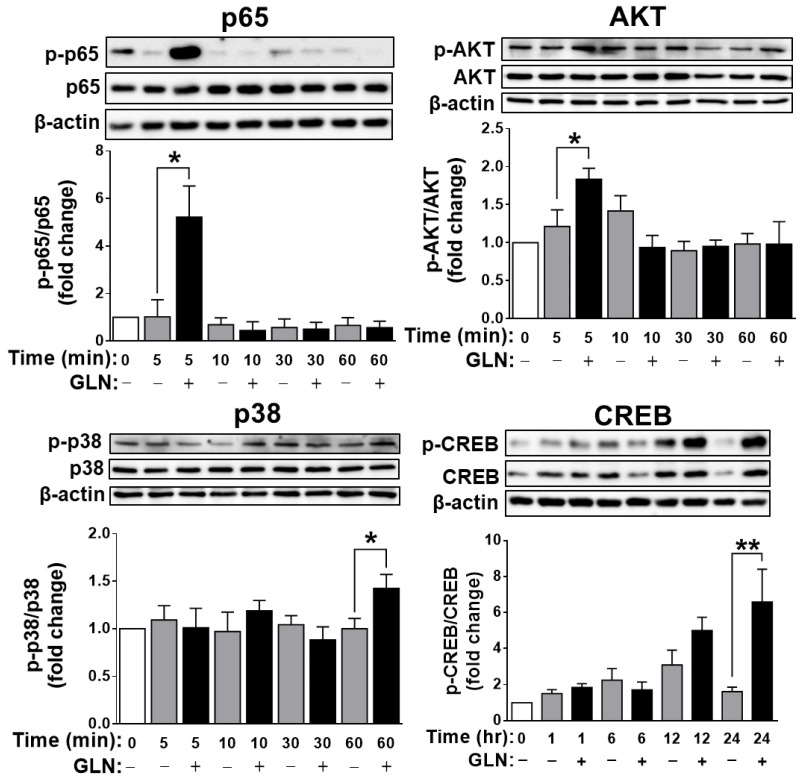
Depiction of the impact of GLN on the activation of signaling molecules in HT22 cells. HT22 cells were serum starved for 1 h before treatment with GLN (20 mM) for specified durations. The phosphorylation pattern of proteins was assessed by Western blotting. The results represent the means ± S.E.M. (*n* = 4) * *p* < 0.05 and ** *p* < 0.01 compared with the 0 mM group at the same time point.

**Figure 7 ijms-25-04211-f007:**
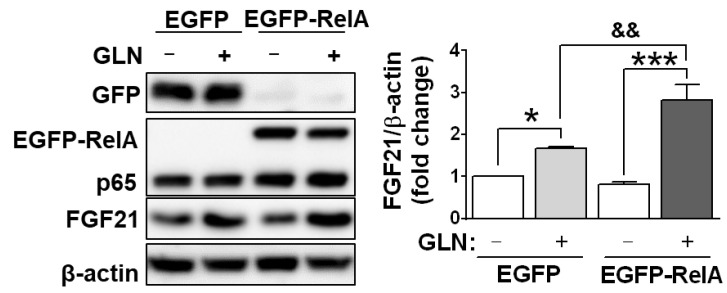
Demonstration of the further induction of FGF21 through overexpression of NF-κB p65 in HT22 cells. HT22 cells were transfected with EGFP-p65 or EGFP plasmid and treated with GLN (20 mM) for 24 h. Western blotting was performed to detect the proteins for FGF21, GFP, EGFP-RelA, and p65 using β-actin as an internal control. The results represent the means ± S.E.M. (*n* = 3). * *p* < 0.05 and *** *p* < 0.001 compared with the 0 mM group. ^&&^ *p* < 0.01 between the EGFP and EGFP-RelA groups.

## Data Availability

All data generated or analyzed during this study are included in this published article.

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
