# Peer review of "Glucosamine Enhancement of Learning and Memory Functions by Promoting Fibroblast Growth Factor 21 Production"

_ijms, 2024, doi:10.3390/ijms25084211_

Round 1
Reviewer 1 Report
Comments and Suggestions for Authors
The article "Glucosamine Enhancement of Learning and Memory Functions by Promoting Fibroblast Growth Factor 21 Production" investigates glucosamine's (GLN) role in cognitive enhancement through FGF21 production in mice. It reveals that GLN significantly improves learning and memory across different diets, linking this effect to increased FGF21 expression in key brain regions and in vitro cell models. The study identifies NF-κB, Akt, p38, and PKA/CREB pathways as crucial for GLN-induced FGF21 expression and highlights the involvement of the FGFR1 receptor in mediating cognitive benefits.
This research provides new insights into the neuroprotective potential of dietary supplements like GLN, suggesting its therapeutic value for cognitive health. By elucidating the molecular mechanisms through which GLN affects FGF21 production and signaling, the study opens avenues for future research on dietary interventions for cognitive impairments and neurodegenerative conditions. This work underscores the importance of dietary components in modulating brain function and presents GLN as a promising agent for enhancing cognitive abilities, offering a foundation for further clinical exploration.
To enhance the quality of your manuscript, the following improvements are recommended:
Introduction
1. The introduction is well-structured, beginning with a broad statement about the importance of learning and memory, followed by the roles of various brain regions and the impact of their degeneration. The transition to discussing FGF21 and GLN is smooth, providing a clear rationale for the study. However, it might be beneficial to briefly explain how FGF21 and GLN specifically relate to learning and memory earlier in the introduction to provide immediate context for their significance in the study.
2. The introduction is well-cited, indicating a solid foundation in the current literature. Ensure that all references are up-to-date and include the most recent and relevant studies, especially concerning FGF21 and GLN's roles in brain physiology and neuroprotection. It might be helpful to include a sentence or two discussing the most current findings or controversies in this area to highlight the study's relevance.
3. The claims made about FGF21 and GLN are broad. It would be beneficial to specify the mechanisms by which FGF21 influences brain physiology and how GLN exerts its neuroprotective effects. Including brief mentions of known pathways or hypotheses could enrich the introduction's depth and provide a clearer direction for the study.
4. While the introduction concludes with the study's aim, it could be strengthened by explicitly stating the hypothesis or research question. This clarification would provide a direct link between the background provided and the study's objectives, making the introduction more compelling and focused.
5. The introduction mentions the global health issue related to learning and memory degeneration but does not elaborate on the potential broader implications of the study's findings. Expanding on how understanding the impact of GLN on FGF21 production in the brain could contribute to therapeutic strategies would enhance the introduction's significance.
6. The introduction uses technical terms appropriately, but consider the target audience's familiarity with terms like "FGF21 cognate receptor FGFR1." A brief explanation or simplification of highly technical terms could make the section more accessible to a broader audience.
Results
7. The experimental design is clearly outlined, with distinct subsections for each major finding. The results are presented in a structured manner, making it easy to follow. However, it could be beneficial to briefly recap the study's purpose at the beginning of this section to remind readers of the context.
8. The use of statistical indicators (e.g., *P<0.05, **P<0.01, ***P<0.001) is appropriate and helps in quickly identifying significant findings. Ensure that the statistical methods used for analysis are described in detail in the methods section, including the statistical tests used and the criteria for significance.
9. The results indicating that GLN enhances learning and memory functions and induces FGF21 expression are compelling. It would strengthen the manuscript to include a discussion on the potential mechanisms by which GLN could be exerting these effects, especially in the context of existing literature. This could be briefly touched upon here and expanded in the discussion section.
10. The results demonstrate a dose-response effect of GLN on FGF21 expression and cognitive functions, which is an important aspect of the study. It could be insightful to discuss the implications of this dose-response relationship further, particularly in determining the optimal dose for therapeutic purposes.
11. It's commendable that the study includes both male and female animals and reports no significant differences in the effects of GLN between genders. This is an important aspect of research that is often overlooked. A brief discussion on the importance of including both sexes in preclinical research and how these findings contribute to understanding potential gender differences in response to GLN could be valuable.
12. While the study's design seems robust, consider if there were any potential confounders or limitations that could have influenced the results. For instance, the impact of the high-fat diet (HFD) on the metabolism of GLN or FGF21 could be worth discussing, as well as any behavioral or environmental variables that were controlled or monitored during the study.
13. The results section concludes with findings that suggest a specific pathway (FGF21) through which GLN enhances cognitive functions. It would be interesting to briefly suggest future research directions that could explore this pathway further, perhaps investigating the molecular mechanisms or the involvement of other cognitive functions or diseases.
14. The choice of signaling pathways (e.g., NF-κB, p38, JNK, Akt, PKA, PPAR) for investigation is relevant and well-justified based on previous studies indicating their involvement in FGF21 expression. It would enhance the manuscript to briefly discuss why these specific pathways were chosen and their known roles in FGF21 regulation.
15. The results indicate a complex regulatory network involving multiple signaling pathways in GLN-induced FGF21 expression. A more detailed discussion on how these pathways might interact or influence each other would be insightful. Additionally, considering the differential effects observed in hippocampal versus striatal cells, elaborating on potential cell-type specific regulatory mechanisms could be interesting.
16. The statistical analysis appears to be thorough, with appropriate significance markers. Ensuring that the statistical methods are consistently applied and clearly explained in the manuscript will aid readers in understanding the robustness of the findings. Also, consider whether any additional statistical tests (e.g., multiple comparison corrections) were necessary given the number of comparisons made.
17. The findings contribute to a better understanding of the molecular mechanisms underlying GLN's effects on FGF21 expression. It would be valuable to speculate on the potential mechanisms by which GLN influences these signaling pathways and, in turn, FGF21 expression. Such discussion could propose hypotheses for future research.
18. The identification of signaling pathways involved in GLN-induced FGF21 expression has broader implications for understanding the molecular basis of GLN's neuroprotective effects. Discussing how these findings contribute to the field and could inform therapeutic strategies for neurodegenerative diseases or cognitive impairment would contextualize the importance of the work.
19. Acknowledging any limitations of the study, such as the exclusive use of in vitro models for some analyses or the potential for off-target effects of the inhibitors used, would provide a balanced view. Additionally, suggesting future studies to validate these findings in vivo or explore the functional consequences of these signaling pathway alterations in the context of GLN's neuroprotective effects would be constructive.
Discussion
20. The discussion effectively summarizes key results, including the enhancement of learning and memory by GLN in mice, the induction of FGF21 in brain regions and cell types, and the identification of signaling pathways involved. It might be beneficial to start this section with a brief statement that directly links these findings to the gaps identified in the introduction, emphasizing how this study advances our understanding.
21. The discussion thoughtfully compares the current findings with previous research, acknowledging both consistencies and discrepancies. It would strengthen the discussion to further explore reasons for these discrepancies, such as differences in experimental design, species used, or doses of GLN and inhibitors. This could help readers appreciate the complexity of GLN's effects on FGF21 production and its cognitive implications.
22. The section on signaling pathways provides a detailed account of the study's novel contributions to understanding GLN's mechanism of action. To further enrich this discussion, consider hypothesizing about the interplay between these pathways based on the findings. For instance, how might the activation of NF-κB, p38, Akt, and PKA/CREB pathways converge to regulate FGF21 expression? Discussing potential cross-talk or feedback mechanisms could offer intriguing directions for future research.
23. While the discussion mentions the potential cognitive benefits of GLN and its safety based on LD50 values, expanding on the translational potential of these findings could enhance the manuscript's impact. Discuss the implications for developing GLN or FGF21-targeted therapies for cognitive disorders, considering factors like dosing, bioavailability, and the need for long-term safety studies.
24. The manuscript acknowledges some limitations and calls for further investigation into the source of FGF21's beneficial effects and the specific signaling interactions. Expanding on these limitations, such as the use of specific animal models or the duration of dietary interventions, and suggesting specific future research directions would provide a roadmap for advancing the field. For example, proposing studies to explore GLN's effects in models of neurodegenerative diseases or detailing how to investigate the signaling pathway interactions would be valuable.
25. Consider situating the findings within a broader context of neuroprotective and regenerative medicine. Discuss how understanding the role of dietary supplements like GLN in modulating neurophysiological processes opens new avenues for preventative strategies or adjunct therapies in cognitive impairment and neurodegeneration.
26. The conclusion succinctly encapsulates the study's contributions but could be enhanced by a forward-looking statement that inspires future research. Highlighting the potential for uncovering novel therapeutic targets or developing innovative treatment strategies based on the study's findings would provide a compelling end to the discussion.
Comments on the Quality of English Language1. It is crucial to maintain consistency in the use of specific terms and nomenclature throughout the manuscript. For example, ensure that abbreviations like GLN for glucosamine and FGF21 for fibroblast growth factor 21 are consistently defined at their first occurrence and then used uniformly throughout the text.
2. While the manuscript is generally well-written, some sentences could be simplified for clarity and conciseness. Avoiding overly complex sentences and using direct language can help in making the manuscript more accessible to a broader audience, including those for whom English might not be their first language.
Reviewer 2 Report
Comments and Suggestions for Authors
REVIEWER COMMENTS
Manuscript Title - Glucosamine Enhancement of Learning and Memory Functions by Promoting Fibroblast Growth Factor 21 Production
Manuscript number - ijms-2883397
The Manuscript needs major revision. The authors need to clarify the composition of the diet and their observation of no change in learning memory after HFD. However, the authors have not provided sufficient justification for this effect, nor have they measured any blood-based parameters for confirmation of obesity. The data should not be split into two graphs for LD and HFD but rather clubbed together for comparison. The manuscript would benefit from proofreading.
Introduction
The introduction provides a good overview of the topic and requires some minor revisions.
- The authors need to correct typos and introduce the effect of glucosamine and FGF21 on glucose and glycogen levels.
- They are encouraged to correlate these with obesity or high-fat content and include an introduction to obesity.
Methods
The methods section requires minor revision The following suggestions are intended to help the authors address the issues:
- Please include a source for the procurement of glucosamine.
- It would be best if the methods for behavior studies were written separately, not with ethics.
- Please write the composition of a normal diet and a high-fat diet in supplementary information.
- Authors are requested to correct the typos for symbols.
- Please mention the timeline for FGFR1 inhibitor PD173074 administration. I am surprised to see that the inhibitor was given concomitantly. How early PD173074 can act? Comment on pharmacokinetics and discuss in method and discussion.
Result
This section needs major revision.
- The authors should compare the results of the normal diet with the high-fat diet in a single graph.
- As per the current figure, the high-fat diet does not cause cognitive deficits. The discrimination ratio in Figure 1 for ND and HFD is around 0.5. There are many articles where 9 or 10 weeks of HFD caused learning and memory deficits (10.1007/s00125-012-2686-y; https://www.frontiersin.org/articles/10.3389/fnbeh.2016.00156/full). May be fat content of the diet was not enough or intake was not monitored. The authors should justify this.
- Please report food intake, glucose tolerance test, and an insulin tolerance test after ND and HFD
- They should also club ND and HFD data in all figures and then do the comparisons using two-way ANOVA. A comparison must be made between ND and HFD, even if there is no difference. Similarly, western blot images should be arranged.
- Additionally, the authors should mention the passage number for cell lines in which experiments were conducted and correct the typos for actin, NFKb, etc.
Discussion
Please provide a discussion about PD173074's fast action. Authors should also discuss the difference in their HFD and others in the field. How fat percentage, time course, and food intake in this study are different from their own previous report (which causes a substantial change in the liver with 8 weeks HFD; https://doi.org/10.1016/j.bbrc.2020.06.070) and others.

Can be improved
Round 2
Reviewer 1 Report
Comments and Suggestions for Authors
I would like to express my sincere gratitude to the authors for their comprehensive and considerate revisions. The manuscript has seen significant improvement from its initial submission, showcasing the authors' commitment to addressing the concerns that were previously mentioned.
Comments on the Quality of English LanguageMinor editing of English language required
Reviewer 2 Report
Comments and Suggestions for Authors
The revision has improved the manuscript. I thank the authors for doing a comprehensive revision. The revised manuscript is recommended for publication.